# Predict the incidence of Guillain Barré Syndrome and arbovirus infection in Mexico, 2014–2019

**Lumumba Arriaga-Nieto[1], Porfirio Felipe Hernández-Bautista[2], Alfonso Vallejos-Parás[1]\*, Concepción Grajales-Muñiz[2], Teresita Rojas-Mendoza[2], David Alejandro Cabrera-Gaytán[3], Israel Grijalva-Otero[4], Bernardo Cacho-Díaz[5], Leticia Jaimes-Betancourt[6], Rosario Padilla-Velazquez[1], Gabriel Valle-Alvarado[1], Yadira Perez-Andrade[1], Oscar David Ovalle-Luna[1], Mónica Rivera-Mahey[1]**

1 Epidemiologic Surveillance Coordination, Instituto Mexicano del Seguro Social, Mexico City, Mexico,
2 Coordination of Supplies Quality Control, Instituto Mexicano del Seguro Social, Mexico City, Mexico,
3 Health Research Coordination, Instituto Mexicano del Seguro Social, Mexico City, Mexico, 4 Medical Research Unit for Neurological Diseases, UMAE Hospital de Especialidades, Centro Médico Nacional Siglo XXI, Instituto Mexicano del Seguro Social, Mexico City, Mexico, 5 Neuroscience Unit. National Cancer Institute, Mexico City, Mexico, 6 Epidemiology Department, Family Medicine Unit 7, Instituto Mexicano del Seguro Social, Mexico City, Mexico

\* alfonso.vallejos@imss.gob.mx

**Data Availability Statement:** All data are available from the public repository database in: https://figshare.com/articles/dataset/Predict_the_

## Abstract

The Dengue (DENV), Zika (ZIKV), and Chikungunya (CHIKV) virus infections have been linked to Guillain-Barré syndrome (GBS). GBS has an estimated lethality of 4% to 8%, even with effective treatment. Mexico is considered a hyperendemic country for DENV due to the circulation of four serotypes, and the ZIKV and CHIKV viruses have also been circulating in the country. The objective of this study was to predict the number of GBS cases in relation to the cumulative incidence of ZIKV / DENV / CHIKV in Mexico from 2014 to 2019. A six-year time series ecological study was carried out from GBS cases registered in the Acute Flaccid Paralysis (AFP) Epidemiological Surveillance System (ESS), and DENV, ZIKV and CHIKV estimated cases from cases registered in the epidemiological vector-borne diseases surveillance system. The results shows that the incidence of GBS in Mexico is positively correlated with DENV and ZIKV. For every 1,000 estimated DENV cases, 1.45 GBS cases occurred on average, and for every 1,000 estimated ZIKV cases, 1.93 GBS cases occurred on average. A negative correlation between GBS and CHIKV estimated cases was found. The increase in the incidence of GBS cases in Mexico can be predicted by observing DENV and ZIKV cases through the epidemiological surveillance systems. These results can be useful in public health by providing the opportunity to improve capacities for the prevention of arbovirus diseases and for the timely procurement of supplies for the treatment of GBS.

## Background

Guillain-Barré syndrome (GBS) is the most common cause of Acute Flaccid Paralysis (AFP) worldwide and is characterized by ascending weakness symmetrical with depression of muscle

incidence_of_Guillain_Barr_Syndrome_and_
Arbovirus_Infection_in_Mexico_2014-2019/
17082470.

**Funding:** The author(s) received no specific
funding for this work.

**Competing interests:** The authors have declared
that no competing interests exist.

stretch reflexes [1]. Approximately 100,000 people develop the disorder each year worldwide [2], and the global incidence of GBS ranges from 0.6 to 4.0/100,000 people [3, 4]. About 20% of patients with GBS develop respiratory failure and require mechanical ventilation [5]. The mortality rate is estimated at 4–8% for patients with GBS even with the best medical care available [6]. In Mexico the hospital mortality rate for GBS has been documented at 10.5% [7]. The long-term impact on a patients life may go beyond their residual disability or impairment, where almost 30% of them have to make substantial changes in their daily lives after suffering from GBS [8].

The underlying causes of the GBS pathology are not yet fully understood [9, 10], However, the following factors have been described as a trigger for GBS: many infections have been linked with GBS, the most common are gastrointestinal or respiratory illnesses. Up to 70% of patients have reported an antecedent illness between one and six weeks before the presentation of GBS [11], about a third of all GBS cases are preceded by *Campylobacter jejuni* infection [12].

Infections by patogens like *Haemophilus influenzae*, influenza virus, Epstein–Barr virus, cytomegalovirus, human immunodeficiency virus, Hepatitis E virus and SARS-CoV-2 are linked with GBS [13–15]; also, several neurological manifestations, including GBS, are associated with Zika (ZIKV), Dengue (DENV) or Chikungunya (CHIKV) viruses' infection [16–21]. Probably the most studied association between GBS and arbovirus infections is ZIKV; however, there are inconsistencies in the studies of the association between ZIKV infection and GBS [22].

Mexico is mainly characterized by a tropical and subtropical climate where dengue is endemic and the number and severity of cases have increased over the last ten years [23, 24]. Mathematical models predict that although the density of mosquitoes is decreasing in Mexico, transmission of dengue will continue because of biological factors that are optimal for the dengue virus [25].

Since laboratory confirmation of the first cases of Zika virus disease in October 2015, and up to 2019, 12,932 autochthonous cases have been identified in 29 of the 32 states of Mexico [26], and 19 cases of GBS associated with Zika have been documented [27]. Since October 2014, when the first autochthonous case of Chikungunya fever was identified and up to 2017, 11,971 Chikungunya cases have been reported in Mexico [28]. A total of 148,883 DENV cases were reported from 2014 to 2019 [29]. The objective of this study is to present a predictive model for GBS cases from DENV, CHIKV and ZIKV cases, arboviruses that circulated simultaneously in Mexico from 2014 to 2019, as proposed by some authors through Epidemiological Surveillance System (ESS) [30]. This study provides information about the association with GBS between DENV, CHIKV and ZIKV infection.

Since 2016, the Epidemiological Surveillance System for Acute Flaccid Paralysis, where the GBS is one of the diagnoses to study, was allowed to report cases in people older than 15 years due to the context of ZIKV, this change in epidemiological surveillance of GBS has been maintained up to date.

All AFP, DENV, ZIKV and CHIKV cases are mandatorily reported to the national ESS in our country. This study was carried out by the Mexican Institute of Social Security (IMSS) data; IMSS is a healthcare system that provides medical assistance to 51% of the Mexican population [31] and as part of the National Health System, also report cases to the Nacional ESS.

## Methodology

Ethics statement: The present study is a part of a larger research protocol of a protocol that was accepted by the IMSS Ethics and Research Commission, Number R-2016-785-026. Since the data were obtained from epidemiological surveillance records, informed consent was not necessary, no procedure was performed outside the norms for epidemiological surveillance including patient samples for laboratory tests. All data analyzed were completely anonymous prior to study.

A population-based six-year time series ecological study was carried out using the AFP cases registered in the ESS from which GBS cases were identified. All cases of AFP are required to be reported to this ESS, and all GBS cases from 2014 to 2019 were included in the study.

The arboviruses cases in the study included all cases of DENV, ZIKV and CHIKV reported from 2014 to 2019, which were obtained by years and epidemiological weeks at the regional level at the Mexican Institute of Social Security.

In Mexico, it is mandatory to report all patients suspected of having DENV, ZIKV and CHIKV infection to the National Epidemiological Surveillance System, and a laboratory sample must be taken to confirm the diagnosis or rule out a case according to the following sampling percentages: a) 30% of non-serious dengue disease cases, 100% of cases with warning signs of severe dengue and 100% of cases with severe dengue disease; b) 10% of all CHIKV reported cases and c) 100% of ZIKV reported cases in pregnant woman and 10% of all ZIKV reported cases in the rest of the population.

## Definitions

GBS: Diagnosis was done according to the clinical criteria of Asbury and Cornblath [32].

The Pan American Health Organization (PAHO) for a suspected arbovirus case definitions were used: the 2009 definitions [33] were used for DENV; the 2016 definition [34] was used for ZIKV; and the 2011 definition [35] was used for CHIKV.

DENV suspected case: defined by a combination of $\geq 2$ clinical findings in a febrile person who traveled to or lives in a dengue-endemic area. Clinical findings include nausea, vomiting, rash, aches and pains, a positive tourniquet test, leukopenia, and the following warning signs: abdominal pain or tenderness, persistent vomiting, clinical fluid accumulation, mucosal bleeding, lethargy, restlessness, and liver enlargement.

ZIKV suspected case: defined as a patient with cutaneous exanthema with two or more of the following signs or symptoms: fever, headache, conjunctivitis (not purulent/hyperemic), arthralgia, pruritus or retroocular pain and any epidemiological association.

CHIKV suspected case: defined as a patient with acute onset of fever and severe arthralgia or arthritis not explained by other medical conditions, and who resides or has visited epidemic or endemic areas within two weeks prior to the onset of symptoms.

A confirmed case of DENV, ZIKV or CHIKV was defined as a suspected case that was determined to be positive by viral RNA detection through real-time RT-PCR in blood serum samples taken within the first five days of clinical symptoms onset (fever or other symptoms).

Estimation of arbovirus cases: laboratory samples were not taken to confirm all DENV, ZIKV or CHIKV suspected cases; therefore, positive cases were estimated according to previous studies [36].

The positivity percent for each disease was determined by dividing the number of positive cases by the sum of positive and negative cases. The positivity was defined as the probability of suffering from DENV, ZIKV and CHIKV infection among people suspected of the disease for whom a laboratory sample was not taken. The positivity was multiplied for suspected cases of DENV, CHIKV, and ZIKV without a laboratory sample, and the result was added to the number of positive cases. The positivity was obtained per epidemiological week and state (region); therefore, the estimated cases were calculated by week and region.

## Data analysis

An epidemic curve was constructed for each individual disease (GBS, DENV, ZIKV and CHIKV) from the sum of the number of cases from all the states of the country for each week from 2014 to 2019. The dengue cases were grouped and added together, regardless of the type

of DENV that was clinically presented (non-severe dengue, dengue with alarm signs or severe dengue). The annual cumulative incidence of each disease was calculated using the population insured by the IMSS at the middle of each year.

To predict GBS and arboviruses, an interrupted time series analysis was carried out through a segmented linear regression [37] using the weekly number of GBS cases as the dependent variable and the weekly arbovirus cases as the independent variables. For linear regression, the DENV, CHIKV, and ZIKV cases were divided by 1,000 to facilitate the interpretation of the results.

The number of weekly cases of GBS and arboviruses were calculated from the date of the onset of symptoms.

The multicollinearity of the variables was analyzed using the variance inflation factor (VIF) and tolerance; VIF values less than 10 and tolerance values greater than 0.10 were used as indicators that the explanatory variables had no multicollinearity in the model [38].

## Results

A total of 1,698 cases of GBS were diagnosed during the studied period, during that time 168,979 DENV, 42,548 ZIKV, and 30,651 CHIKV cases were estimated (Table 1).

The GBS cases increased from 2014 to 2019 by 2.4 times, and the highest growth occurred from 2015 to 2016; The cases and incidence of DENV per year presented changes, both of increase and decrease. In 2019 there was a DENV outbreak in Mexico. The behavior of dengue fever in Mexico has been dynamic over the years, and the outbreak patterns coincide with those factors presents at the regional level [23, 39, 40], among the factors involved is the phenomenon of global warming which has expanded the vector´s habitat, the poor treatment of bodies of water that favors their growth and increases in mosquito density, migratory phenomena and human movements, which together increase the transmission possibilities [41–43].

While the epidemiological behavior has been different for CHIKV and ZIKV diseases, incidences decreased over time after the Chikungunya national outbreak in 2015 and the Zika national outbreak in 2016, respectively, the years when the highest number of cases were recorded Fig 1.

In Fig 1, the average mean of GBS cases before 2016 were 2.93 cases and after that year, 6.68 cases, with an average weekly increase of 77.8%.

Given that the three arboviruses are transmitted by the same vector and therefore circulate at the same time in Mexico, a correlation analysis was performed for these diseases and GBS: the results showed a positive correlation for CHIKV with DENV, DENV with ZIKV, DENV with GBS and ZIKV with GBS, and a negative correlation for CHIKV with GBS.

The results from the collinearity analyses performed for the three arboviruses indicated that a linear regression could be used to correlate arboviruses with GBS.

Segmented linear regression showed that GBS cases were related to the arboviruses with an adjusted R-squared value of 0.408 and a statistical significance of <0.0001. (Table 2).

**Table 1. Number of cases and cumulative incidence rate[*] of GBS, DENV, CHIKV and ZIKV, Mexico IMSS 2014–2019.**

| Disease | 2014 | | 2015 | | 2016 | | 2017 | | 2018 | | 2019 | |
|---|---|---|---|---|---|---|---|---|---|---|---|---|
| | Cases | CIR | Cases | CIR | Cases | CIR | Cases | CIR | Cases | CIR | Cases | CIR |
| GBS | 181 | 0.42 | 127 | 0.29 | 342 | 0.78 | 306 | 0.64 | 300 | 0.61 | 442 | 0.87 |
| DENV | 30,684 | 70.80 | 30,839 | 69.76 | 19,913 | 45.13 | 16,467 | 34.68 | 11,065 | 22.48 | 60,011 | 117.59 |
| ZIKV | 0 | 0.00 | 20 | 0.05 | 35,584 | 80.65 | 6,288 | 13.24 | 610 | 1.24 | 46 | 0.09 |
| CHIKV | 44 | 0.10 | 29,792 | 67.39 | 764 | 1.73 | 31 | 0.07 | 19 | 0.04 | 1 | 0.00 |

[*]CIR = Cumulative incidence rate by 100,000 people.

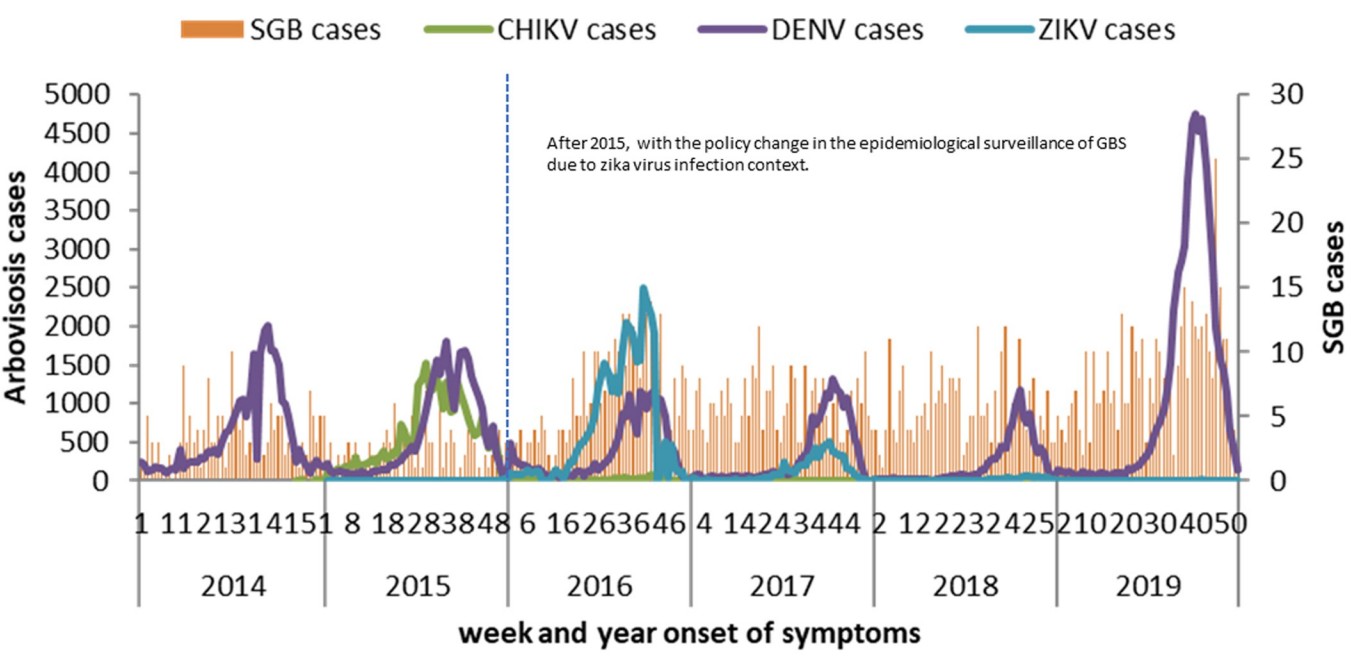

GBS= Guillain-Barré syndrome
CHIKV= Chikungunya virus
DENV= Dengue virus
ZIKV= Zika virus

**Fig 1. Epidemic curve of GBS and arbovirus cases in Mexico, IMSS 2014–2019.**

Because we used a time-segmented model of GBS cases, we have two equations that can be used to estimate the number of GBS cases from the number of arbovirus cases and significant coefficients for the estimated cases of DENV, ZIKV, and CHIKV:

1. Number of GBS cases before 2016 = 2.5 + (1.45 * (estimated number of DEN cases*1,000)) + (1.93 * (estimated number of ZVD cases*1,000))—(1.46 * (estimated number of CHIKV cases*1,000)).

2. Number of GBS cases since 2016 = 5.5 + (1.45 * (estimated number of DEN cases*1,000)) + (1.93 * (estimated number of ZVD cases*1,000))—(1.46 * (estimated number of CHIKV cases*1,000)).

In the equations, the constant is 2.5 cases/week before 2016 and 5.5 cases/week since 2016, the change in the notification policy in 2016 increased GBS notification by 3 cases/week, therefore, it must be considered a βo value equal a 5.5 in the equation for that period (2016 to 2019).

**Table 2. Interrupted time series analysis, segmented linear regression of GBS cases and arboviruses cases, 2014–2019.**

| | Non-standardized coefficients | | Standardized coefficients | Sig. | 95% confidence interval for B | | Collinearity statistics | |
|---|---|---|---|---|---|---|---|---|
| | B | Error Desv. | Beta | | Lower limit | Upper limit | Tolerance | VIF |
| (Constat) | 2.496 | 0.331 | | <0.0001 | 1.845 | 3.147 | | |
| CHIKV/1000 | -1.457 | 0.650 | -0.113 | 0.026 | -2.736 | -0.179 | 0.742 | 1.349 |
| DENV/1000 | 1.452 | 0.197 | 0.328 | <0.0001 | 1.065 | 1.840 | 0.963 | 1.038 |
| ZIKV/1000 | 1.925 | 0.393 | 0.220 | <0.0001 | 1.152 | 2.698 | 0.938 | 1.067 |
| Since 2016 year | 3.048 | 0.379 | 0.411 | <0.0001 | 2.302 | 3.795 | 0.727 | 1.376 |

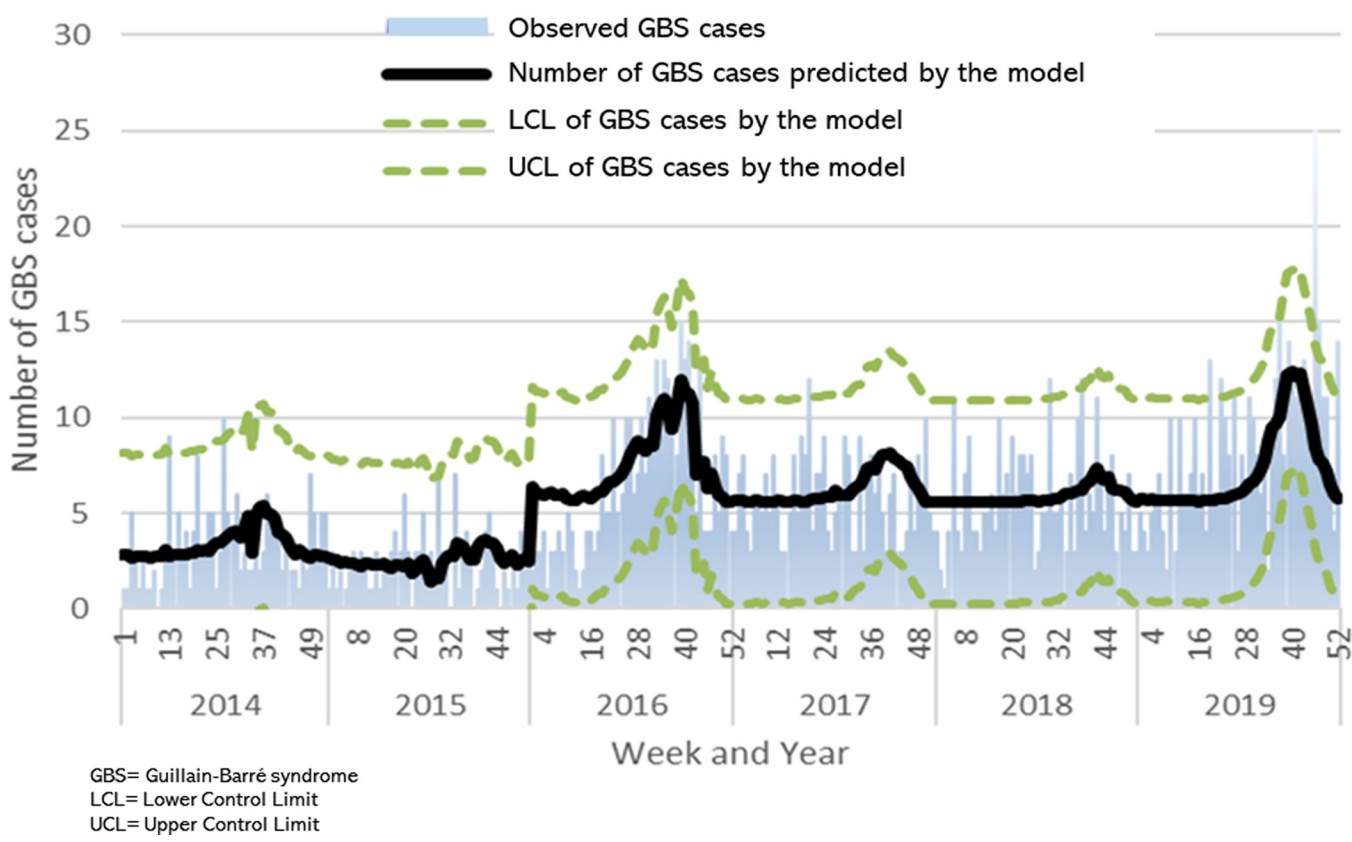

**Fig 2. Epidemic curve of GBS model observed and predicted cases and arbovirus cases in Mexico, IMSS 2014–2019.**

GBS= Guillain-Barré syndrome
LCL= Lower Control Limit
UCL= Upper Control Limit

The interpretation of the results is that in the period studied there were 2.5 cases of GBS every week before the year 2016 and 5.5 cases of GBS every week since 2016, that were not explained by the DENV, CHIKV and ZIKV estimated cases.

GBS cases increase weekly on average by 1.45 per 1000 estimated cases of DENV; They also increase on average by 1.93 cases per 1000 estimated cases of ZIKV; and decrease on average by 1.46 cases per 1000 estimated cases of CHIKV.

Fig 2 shows the cases predicted by the linear regression model with an R squared = 0.415 and statistical significance of p = 0.000 through the Ljung-Box Q test.

When finding a negative result for CHIKV in the model, we decided to perform a segmented linear regression only using the independent variable CHIKV, as a GBS dependent variable and using as segmentation time before the year 2016, which as can be seen in the graphical epidemic curve (Fig 1), is the period where more cases of CHIKV were reported, finding the following:

R squared = 0.016, B = -0.589 (-1.499–0.321), p = 0.202 not statistically significant. This finding is consistent with the global model presented where the correlation of CHIKV and SGB was negative.

## Discussion

A population-based study was performed where the number of DENV, CHIKV and ZIKV cases were correlated with the number of GBS cases to predict the incidence of GBS cases. In our results the estimated incidence of GBS associated to ZIKV cases is 10 times greater to that

published in an ecological study on the American continent, where 2 (0.5–4.5) GBS cases were estimated for every 10,000 ZIKV cases; Nevertheless, the frequency of reported suspect Zika cases varied substantially and the notification was highly uncertain in that study [44]. The relationship between ZIKV infection and GBS is probably the arbovirus infection most studied since the WHO concluded in 2016 that ZIKV infection is a trigger for GBS [45]. In our study, the incidence of GBS increased from 0.3 to 0.8 per 100,000 inhabitants/year from 2015 to 2016, which is equivalent to a 2.6-fold increase. Other Latin American countries (LAC) reported an increase in the cumulative incidence of GBS during the ZIKV epidemic; the incidence of GBS increased 4.4 times in Martinique from 2.12 to 9.35 per 100,000 and by 2.1 times in Puerto Rico from 1.7 to 3.5 per 100,000 inhabitants [46]. In LAC, the incidence of GBS increased 2.6-fold during ZIKV flare-ups and 1.9-fold during CHIKV flare-ups compared to previously reported incidence rates.

An ecological study carried out in Brazil revealed that a rapid increase in the number of hospitalizations for GBS followed the introduction of ZIKV and CHIKV in 2015, where the average number of hospitalizations increased by 45% compared to that from 2008 to 2014 [47]. In another more complex case-control study in Mexico, an association was found between acute ZIKV infection and GBS with an odds ratio of 8.04 (95% CI, 0.89–73.01); p = 0.047 [24]. However, a meta-analysis revealed that ZIKV outbreaks were not consecutive to GBS outbreaks; because only one analytical study included in the analysis indicated an association between the two types of outbreaks, the result for the association in the meta-analysis was 1.57 (95% CI 0.56–2.86) [22].

The association between DENV infection and GBS is rare [24], and it has been estimated that less than 10% of dengue cases develop neurological disorders, including GBS [48, 49]. From our results, we have estimated that for every 1,000 cases of DENV, 1.45 cases of GBS increases. The yearly distribution of GBS and DENV are not the same; specifically, GBS has a low annual frequency and is most prevalent during summer, whereas DENV has a high annual frequency. An ecological study conducted in Hong Kong found a weak correlation between GBS and meteorological factors (17%) [50].

Studies have related the presence of IgM antibodies to CHIKV with GBS [17]; however, it has been difficult to quantitatively estimate the incidence of CHIKV infection with neurological diseases. In the present study, no positive relationship was detected between CHIKV infection and GBS; The study of the relationship between Chikungunya and GBS has been complex, since, in the populations where it has been studied, there exist endemic circulation of DENV, ZIKV and CHIKV. The use of IgM and IgG antibodies reflects a previous infection with CHIKV, a complexity resulting from their transmission dynamics, and more integrative studies are needed investigating the combination of ZIKV, DENV, and CHIKV viruses and using a variety of approaches to answer questions related to the risk posed by these arboviruses [51]. In our study we include symptomatic cases of CHIKV; asymptomatic cases can be at least 47%, more than those included in this study [52]. In the studies that have used PCR to detect CHIKV infection, they have had a low number of samples to associate CHIKV and GBS [53]

Our model indicates that, before 2016 there were 2.5 cases of GBS per week and, from 2016, 5.5 cases of GBS per week that do not correlate to arboviruses; this indicates that there are different etiologies of GBS, probably the most studied infection by *Campylobacter jejuni* infection [12] and there is evidence that GBS cases in the pediatric population in Mexico have a relationship with this bacterium [54].

The prediction of a mathematical model indicate that even with a decrease in the density of mosquitoes in Mexico, transmission will continue because of optimal biological factors for DENV [25]. This model could be used to estimate the number of GBS cases from the burden of DENV and other vector-borne diseases.

The following limitations of the study should be considered: 1) The study was not designed to measure a causal relationship; due to the nature of the data, an exposure-disease relationship cannot be assumed, where the exposure must precede the onset of the disease [55].

Neither analyze demographic factors, such as the age or sex of the patients, although it has been shown that older patients and males have a higher incidence of GBS. 2) This is an ecological study, so its interpretation should not be at the individual level in order to not to fall into an ecological fallacy. 3) Note that coinfection between arboviruses was not analyzed, given that arbovirus cases were analyzed independently of each other; However, it is not yet known if the coinfection impacts clinical disease [56], but it has been described that GBS patients with laboratory evidence of both a recent ZIKV and CHIKV infection had a longer duration of hospitalization, were admitted to the ICU, and intubated significantly more frequently than the other patients with GBS [21].

4) The GBS clinical variant was not considered in this study, all cases regardless of the type of variant were included in the analysis. The clinical phenotype of GBS associated with ZIKV infection reported in literature is generally a sensorimotor demyelinating GBS with frequent facial palsy and a severe disease course often necessitating ICU admittance [57].

5) A surveillance bias exists because surveillance of GBS was reinforced by the identification of autochthonous cases of ZIKV by allowing the notification of GBS in people older than 15 years, since it was very specific for the search for flaccid paralysis in search of poliomyelitis.

## Conclusions

Our results in this study indicate a positive relationship between ZIKV and DENV with GBS; thus, in the face of abrupt increases in these diseases, it is recommended that incidences of GBS in Mexico be monitored and the health system be prepared to care for these cases.

## Author Contributions

**Conceptualization:** Lumumba Arriaga-Nieto, Porfirio Felipe Hernández-Bautista, Alfonso Vallejos-Parás, Israel Grijalva-Otero.

**Data curation:** Lumumba Arriaga-Nieto, Porfirio Felipe Hernández-Bautista, Alfonso Vallejos-Parás, Leticia Jaimes-Betancourt, Gabriel Valle-Alvarado, Oscar David Ovalle-Luna, Mónica Rivera-Mahey.

**Formal analysis:** Lumumba Arriaga-Nieto, Alfonso Vallejos-Parás, Leticia Jaimes-Betancourt, Mónica Rivera-Mahey.

**Investigation:** Porfirio Felipe Hernández-Bautista, Concepción Grajales-Muñiz, Teresita Rojas-Mendoza, Israel Grijalva-Otero, Leticia Jaimes-Betancourt.

**Methodology:** Porfirio Felipe Hernández-Bautista, Alfonso Vallejos-Parás, Teresita Rojas-Mendoza, Israel Grijalva-Otero, Bernardo Cacho-Díaz, Gabriel Valle-Alvarado, Yadira Perez-Andrade, Oscar David Ovalle-Luna.

**Supervision:** Alfonso Vallejos-Parás, Rosario Padilla-Velazquez.

**Validation:** Alfonso Vallejos-Parás, Concepción Grajales-Muñiz, David Alejandro Cabrera-Gaytán, Rosario Padilla-Velazquez, Gabriel Valle-Alvarado, Yadira Perez-Andrade, Mónica Rivera-Mahey.

**Visualization:** Yadira Perez-Andrade.

**Writing – original draft:** Alfonso Vallejos-Parás, Bernardo Cacho-Díaz.

**Writing – review & editing:** Alfonso Vallejos-Parás, David Alejandro Cabrera-Gaytán, Bernardo Cacho-Díaz, Rosario Padilla-Velazquez.

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
