## [Decision Letter · Decision Letter 0]

29 Sep 2021

PGPH-D-21-00407

Temporal Correlation Between Guillain Barré Syndrome and Arbovirus Infection in Mexico, 2014-2019.

Dear Dr. Vallejos-Parás,

Thank you for submitting your manuscript to PLOS Global Public Health. After careful consideration, we feel that it has merit but does not fully meet PLOS Global Public Health’s publication criteria as it currently stands. Therefore, we invite you to submit a revised version of the manuscript that addresses the points raised during the review process.

We look forward to receiving your revised manuscript.

Kind regards,

Mathieu Nacher

Academic Editor

Journal Requirements:

1. In your ethics statement in the manuscript, please ensure that you have discussed whether all data/samples were fully anonymized before you accessed them and/or whether the IRB or ethics committee waived the requirement for informed consent. If patients provided informed written consent to have data/samples from their medical records used in research, please include this information.

2. Please provide  separate figure files in .tif or .eps format only.  Please ensure that all files are under our size limit of 20MB.  

For more information about how to convert your figure files please see our guidelines: Once you've converted your files to .tif or .eps, please also make sure that your figures meet our format requirements

3. We have noticed that you have uploaded supporting information but you have not included a list of legends.  Please add a full list of legends for all supporting information files (including figures, table and data files) after the references list. 

4. In the online submission form, you indicated that your data will be submitted to a repository upon acceptance.  We strongly recommend all authors deposit their data before acceptance, as the process can be lengthy and hold up publication timelines. Please note that, though access restrictions are acceptable now, your entire data will need to be made freely accessible if your manuscript is accepted for publication. This policy applies to all data except where public deposition would breach compliance with the protocol approved by your research ethics board. If you are unable to adhere to our open data policy, please kindly revise your statement to explain your reasoning and we will seek the editor's input on an exemption. Please be assured that, once you have provided your new statement, the assessment of your exemption will not hold up the peer review process.

Additional Editor Comments (if provided):

Reviewers' comments:

Reviewer's Responses to Questions

**Comments to the Author**

1. Does this manuscript meet PLOS Global Public Health’s publication criteria? Is the manuscript technically sound, and do the data support the conclusions? The manuscript must describe methodologically and ethically rigorous research with conclusions that are appropriately drawn based on the data presented.

Reviewer #1: Partly

Reviewer #2: Yes

2. Has the statistical analysis been performed appropriately and rigorously?

Reviewer #1: Yes

Reviewer #2: I don't know

3. Have the authors made all data underlying the findings in their manuscript fully available (please refer to the Data Availability Statement at the start of the manuscript PDF file)?

Reviewer #1: No

Reviewer #2: No

4. Is the manuscript presented in an intelligible fashion and written in standard English?

Reviewer #1: No

Reviewer #2: No

5. Review Comments to the Author

Reviewer #1: The main objective of this paper is predict the incidence of Guillain-Barré Syndrome in Mexico from the incidence of Dengue, Zika and Chikungunya diseases using a linear regression model applied to national surveillance data collected between 2014 and 2019. These three arboviroses already represent a heavy burden for the population of Latin America, GBS is an added complication for both patients and health systems. The data and analysis presented in this paper is of relevance to a public health audience.

* Title and abstract

The title of the paper and the objective section of the abstract do not reflect the main objective of the study which is to predict the incidence of GBS, of public health interest, whereas correlations appear secondary when reading the whole manuscript.

The methods section of the abstract mentions a five-year analysis when a six-year analysis is presented in the paper (2014-2019).

In the results section: “The increase of incidence of GBS in Mexico is correlated with the epidemic years of DENV and ZIKV”. None of the results of the paper supports this particular claim. The linear regression or the correlations shown in the paper do not allow drawing conclusions that distinguish epidemic vs. non-epidemic years.

* Background

This introduction does not mention the different etiologies of GBS other than arboviruses (eg: Campylobacter infections). Surprisingly, the lethality figure mentioned in the abstract is not found in the background section; with a proper reference, this figure would help stressing out the importance of GBS as a health problem.

Line 73: The formulation of the objective is too vague. In my opinion the interest of the paper is the presentation of a predictive model and should be focused on it. This should also be reflected in the title of the paper.

* Methods

Case definitions and sources of data are well described.

Linear regression: Line 127 “to correlate”; the term should be changed. The objective of the model is not to study correlations but to predict the number of GBS cases for each increase in the incidence of ZKV/DENV/CHIKV. Please precise that the dependent variable is also the weekly number of GBS cases (line 127).

Please also clarify what is the date used for the cases: date of symptom onset, date of reporting?

ANOVA: Why was an ANOVA performed? It is unclear what variables were compared in the ANOVA: “weekly means and annual GBS cases”; is the variance of weekly cases of GBS compared between each year? Why?

* Results

Table 1

Please harmonize acronyms in Table 1 with the rest of the text: SGB vs GBS, DEN vs DENV, etc. The first year should be 2014 and not 2015.

There is clearly a change in the yearly number of cases of GBS before and after 2016. As mentioned in the discussion, this could be attributed to a change in the reporting policy especially in the context of ZKV epidemic and international awareness. This could be estimated with an interrupted time series analysis (ITSA, segmented linear regression) by adding to the linear regression model a step variable (0 before 2016, 1 after) to model this change in the surveillance system. The associated coefficient would estimate the before/after variation.

Line 145: I find a lack of epidemiological rigor in the way the evolution of yearly cases of GBS and arboviruses cases is presented. Especially: line 146: “the incidence and number of DEN cases doubled from 2015 and 2019” is quite misleading. Graph 1 clearly shows that each year Mexico experiences a DENV epidemic, the magnitude of which appears to decrease slightly between 2014 and 2018. Only in 2019 the epidemic was particularly big. Do the authors have any explanation for this observation?

Table2

The ANOVA shows that the yearly mean of weekly GBS cases are statistically different over the period, this result in itself is not much more informative than those shown in Table 1 and Graph 1. It clearly shows an increase between 2014-2015 and 2016-2019; another argument in favor of a change of reporting policy.

Line 168: why is this statement important? For a linear regression, only the distribution of error terms should be normal. Was this checked?

Line 163/Line 171: please be clearer about why despite positive correlations between independent variables only limited collinearity was found. The correlations presented in the appendix do not bring much added information.

Table 3

The values shown for the coefficients and associated 95%CI have too few significant numbers. Maybe a change in the unit of the independent variables (i.e.: 1000 cases instead of single case) would make the table and the equation more readable. It would also better fit the interpretation given in the discussion (lines 187 and 188).

Have you tried a simpler model using a single independent variable grouping all cases of arboviruses instead of distinguishing the diseases?

Again, I suggest trying an ITSA to explore the effect of change of policy in the reporting of GBS.

* Discussion

The constant of the model (4.2 to 5.1 cases of GBS per week in absence of arboviroses) should be discussed. So should the negative coefficient for CHIKV (cf lines 216-218).

Line 188: “This finding is very similar”, the ZIKV-associated incidence of GBS estimated in this article is 10 times greater than the mean value found in ref #21. I don’t think ‘very similar’ is appropriate.

Lines 225-228: This is unclear. What is meant by a “temporality bias”? Indeed a time lag exists between ZIKV infection and GBS onset: was it taken into account into the analysis? If not, why?

Lines 235-236: There is a clear shift before and after 2016 in the total number of GBS cases reported each year, despite the absence of Zika epidemic after 2016. A change in surveillance and reporting of GBS is a major bias and should be explored before discussing trend evolution between 2014 and 2019.

Line 233: “infected with GBS”: this phrasing is not appropriate; GBS is an auto-immune disorder that may result from an infection, but not an infectious disease in itself. Did the authors write GBS instead of ZKV?

Line 238: “DENV” should be replaced by “GBS”

Reviewer #2: Cumulative data support an association between ZIKV and GBS in some parts of the world. However, uncertainty remains regarding the association between other arbovirus cases and GBS.

The authors conducted a population-based study analysing a 6-year time series in Mexico (2014-2019). They assessed the correlation between Guillain-Barré syndrome cases and arboviruses cases (DENV, ZIKV, CHIKV).

The data are based on a nationwide epidemiological surveillance system of both GBS and arbovirus cases.

The methods are well explained and the results are clearly presented. The identification of arbovirus cases is well explained and clear. The gathering of data is well designed.

The authors conclude that GBS cases are positively correlated with DENV and ZIKV cases and determined the strenght of this association. On the opposite they report a negative correlation between CHIKV and GBS cases which is a matter of debate.

The authors also discuss thoroughly the limitations of their study.

The data seem robust to support the conclusions of the authors.

The manuscript is quite concise and clear and as a significant potential for publication.

However I have some remarks regarding the manuscript.

First, the statistical analysis plan need to be assessed by a times series analysis specialist to check if.

Second, the authors should strengthen what this study adds to the current knowledge about the subject. I suggest the introduction section should state more clearly what is already known about it and what remains to be studied.

Even though the manuscrit is in general comprehensive some language editing should made throughout the manuscript to gain in clarity (ex l 94-95, 118, 121-122, l 204)

Some of the results are given in the discussion section(see L 188-189) (or the abstract "For every 1,000 cases of dengue, 2 cases of GBS occurred, and for every 1,000 cases of Zika, 3 cases of GBS occurred") and should be cited in the results section because it corresponds to the main objective of the study.

L 231 231 the authors state that "there is no evidence that coinfection between DENV and ZIKV increases the risk of GBS". This point has been assessed somewhere else and should be discussed see Leonhard et al https://doi.org/10.1016/j.jns.2020.117272

The authors find a negative correlation between CHIK and GBS cases. Tis point is not really discussed after. It could be interesting because some authors found a possible association between CHIKV and GBS (see also Leonhard et al).

The time bias evoked by the authors should be discussed more thoroughly. Indeed this point limits the interpretation of the temporal correlation found here. Is the short timelapse (6-11 days) cited by the authors between arbovirus infection and GBS occurrence is really that certain?

The discussion section could be somewhat reviewed to clarify the authors’ point.

L 211 Do the authors refer to their result or to the literature?

Table 2 Do the authors refer to ANOVA of weekly means?

In the abstract the authors wrote 5-year time series whereas it is 6-year time series.

6. PLOS authors have the option to publish the peer review history of their article (what does this mean?). If published, this will include your full peer review and any attached files.

**Do you want your identity to be public for this peer review?** For information about this choice, including consent withdrawal, please see our Privacy Policy.

Reviewer #1: No

Reviewer #2: **Yes: **Paul Le Turnier

---

## [Decision Letter · Decision Letter 1]

7 Jan 2022

PGPH-D-21-00407R1

Predict the incidence of Guillain Barré Syndrome and Arbovirus infection in Mexico, 2014-2019

Dear Dr. Vallejos-Parás,

Thank you for submitting your manuscript to PLOS Global Public Health. After careful consideration, we feel that it has merit but does not fully meet PLOS Global Public Health’s publication criteria as it currently stands. Therefore, we invite you to submit a revised version of the manuscript that addresses the points raised during the review process.

We look forward to receiving your revised manuscript.

Kind regards,

Mathieu Nacher

Academic Editor

Journal Requirements:

Additional Editor Comments (if provided):

Reviewers' comments:

Reviewer's Responses to Questions

**Comments to the Author**

1. If the authors have adequately addressed your comments raised in a previous round of review and you feel that this manuscript is now acceptable for publication, you may indicate that here to bypass the “Comments to the Author” section, enter your conflict of interest statement in the “Confidential to Editor” section, and submit your "Accept" recommendation.

Reviewer #1: (No Response)

Reviewer #2: All comments have been addressed

2. Does this manuscript meet PLOS Global Public Health’s publication criteria? Is the manuscript technically sound, and do the data support the conclusions? The manuscript must describe methodologically and ethically rigorous research with conclusions that are appropriately drawn based on the data presented.

Reviewer #1: Yes

Reviewer #2: Partly

3. Has the statistical analysis been performed appropriately and rigorously?

Reviewer #1: Yes

Reviewer #2: I don't know

4. Have the authors made all data underlying the findings in their manuscript fully available (please refer to the Data Availability Statement at the start of the manuscript PDF file)?

Reviewer #1: Yes

Reviewer #2: Yes

5. Is the manuscript presented in an intelligible fashion and written in standard English?

Reviewer #1: No

Reviewer #2: No

6. Review Comments to the Author

Reviewer #1: Thank you for addressing the main concerns raised during the first round of review.

General comments:

I particularly appreciate the additions brought to the introduction, and the author's effort to conduct an interrupted time series analysis. This seems a genuine improvement.

However I think the English of the manuscript needs additional proofreading before publication.

Specific comments:

- Graph 2 displays the same information as Graph 1 with a scale change and highlight of 2016 policy change. I suggest combining the two figures, especially given Graph 3 displays actual vs. modelled data.

- Line 211: the effect of 2016 policy change is missing from the equation of the model

- Line 217: the interpretation is not correct; after 2016 the notification increases by three (additional model) but is not multiplied by 3. The constant is 2.5 cases/week before 2016 and 5.5 cases/week after 2016.

- I suggest avoiding reporting zero p-values (i.e. p=0.000 in Table 1) and using p<0.0001 instead.

Reviewer #2: The authors have made significant efforts to answer the questions of both reviewers and have improved the manuscript. The conclusions are supported by the data. However, a language editing remains necessary to gain in clarity in many parts of the manuscript considering some authors' points are still hard to catch (ex l223-226, 264-270).

7. PLOS authors have the option to publish the peer review history of their article (what does this mean?). If published, this will include your full peer review and any attached files.

**Do you want your identity to be public for this peer review?** For information about this choice, including consent withdrawal, please see our Privacy Policy.

Reviewer #1: **Yes: **Yann Lambert

Reviewer #2: **Yes: **Paul Le Turnier

---

## [Editor Report · Decision Letter 2]

2 Mar 2022

Predict the incidence of Guillain Barré Syndrome and Arbovirus infection in Mexico, 2014-2019

PGPH-D-21-00407R2

Dear Mr Vallejos-Parás,

We are pleased to inform you that your manuscript 'Predict the incidence of Guillain Barré Syndrome and Arbovirus infection in Mexico, 2014-2019' has been provisionally accepted for publication in PLOS Global Public Health.

Best regards,

Mathieu Nacher

Academic Editor